# Natural Language Annotations for Reasoning about Program Semantics

**Marco Zocca**

UnfoldML, Göteborg, Sweden
marco.zocca@unfoldml.com

## Abstract

By grounding natural language inference in code (and vice versa), researchers aim to create programming assistants that explain their work, are "coachable" and can surface any gaps in their reasoning. Can we deduce automatically interesting properties of programs from their syntax and common-sense annotations alone, without resorting to static analysis? How much of program logic and behaviour can be captured in natural language? To stimulate research in this direction and attempt to answer these questions we propose HTL , a dataset and protocol for annotating programs with natural language predicates at a finer granularity than code comments and without relying on internal compiler representations.

The dataset is available at the following address: https://doi.org/10.5281/zenodo.7893113 .

## 1 Introduction

Software engineering is an interactive, social practice, in which communication with peers and reading program code are as important if not more than the act of writing code itself. Recent focus on the application of large language models to programming enabled the analysis and synthesis of ever-growing segments of text (both for program synthesis (Xu et al., 2022) and summarization (Wang et al., 2021)), with helpful and remarkably human-like responses, hinting at the possibility of artificial assistants to augment human users.

A common issue with generative models of text is that their output cannot be explicitly controlled, and as such it may contain factual inaccuracies, copies of the training set, etc. It is desirable, then, that such models produce explanations of their output, in the form of "entailment trees"(Dalvi et al., 2021), references to their sources, and the like.

Inspired by recent results in explainable open-domain question answering (Dalvi et al., 2021; Tafjord et al., 2021) and natural language inference

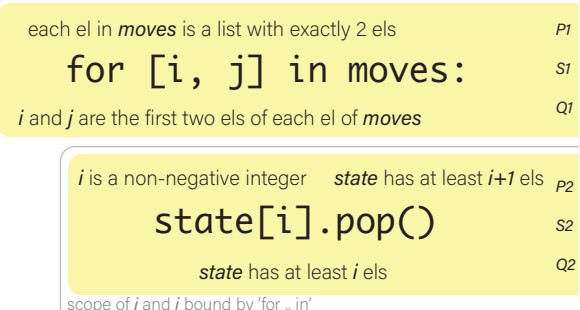

Figure 1: Two program statements $S_1$ and $S_2$ annotated with one or more pre-and post-conditions. Can an automatic system deduce $Q_2$ or $P_1$ from the remaining annotations?

with missing premises (Sprague et al., 2022) we ask whether it is possible to *reason about programs with natural language*.

As a first step towards answering this question, we present a bi-modal dataset, HTL ("Hoare Triples via Language"), composed of natural language predicates describing a pre- and post-condition of a given program expression or statement, paired with spans of program code.

By asking practitioners to produce behavioural annotations of statements and larger blocks of code as they would communicate to a mentee, we aim to capture "intuitive theories" (Tenenbaum et al., 2007) of program semantics in language, i.e. sketches of causal knowledge that can guide deduction and inform explanation of results.

A natural language approximation of program semantics aims to fill the gap between code "comments" (as programmers usually attach to functions or module headers) and exact formal methods. On one hand, comments can be highly informative about programmer intent (which may or may not coincide with program behaviour) but generally follow no set quality standard and contain jargon and abbreviations; on the other, formal methods such as static analysis and symbolic execution provide sound approximations but require expertise in for-

```
for v, f in zip(valids, filenames):
    n_digits = sum(c.isdigit() for c in f)
```

```
P : "'c' is a character"
S : c.isdigit()
Q : "'true' if 'c' contains a digit"
```

```
P : "'f' is a string"
S : c.isdigit() for c in f
Q : "a list with 'true' at each position where 'f' contains a digit character and
    ↪ 'false' otherwise"
```

```
P : "'filenames' is a list of strings"
S : for v, f in zip(valids, filenames):
Q : "'f' is bound to each string in 'filenames'"
```

Figure 2: A code snippet from Schuster et al. (2021) and three HTL labeled instances obtained from it. The annotation triples are presented in order of inclusion of the variables' lexical scope, see the Annotation Guidelines in Section 2 for details. Please refer to Appendix 5 for a full description of the schema.

malizing the specification (Iordache and Ciobaca, 2021) as well as advanced tooling, when implementations even exist.

To summarize, in this paper we make the following contributions:

- We introduce HTL , a new dataset of fine-grained natural-language annotations of program behaviour

- We describe the annotation protocol and tool we developed to produce the dataset.

**Background**  Hoare (1969) laid an axiomatic foundation for proving properties of computer programs, by introducing *triples* (written as $\{P\}\,S\,\{Q\}$ ) relating a precondition predicate $P$, a statement $S$ and a postcondition predicate $Q$ that results from executing the statement (provided this terminates). It is worth noting that predicates $P$ and $Q$ can apply either to individual variables in the program or to sets of those. The Hoare formalism has been extended in many ways since the original publication to account for features of modern programming languages such as lexical scope and recursive function calls (de Boer and Hiep, 2021). By relating pairs of triples with appropriate rules, Hoare logic deduces pre- and postconditions of larger programs; for example, in order to compute the final postcondition of a two-statement program $S_1;S_2$ , the composition axiom of Hoare triples can then be invoked, as long as the intermediate postcondition coincides with or implies the next precondition: in symbols, given $\{P_1\}\,S_1\,\{Q\}$

and $\{Q\}\,S_2\,\{R\}$ , the compound triple is written $\{P_1\}\,S_1;S_2\,\{R\}$ . In the example of Figure 1, the two triples can be composed because the second one appears in the lexical scope introduced by the first one.

## 2   The HTL dataset

**Desiderata**  One of our aims while preparing this dataset was that the "code" part should be straightforward to verify. The source data should then be made of code that is *self-contained* (i.e. needs no external libraries) and it should be *efficiently computable* as a ground truth. In addition, it should be possible to *reason locally* about the behavior of the statements to be annotated; this restricted us to an idealized fragment of the programming language, e.g. without object class declarations and instances.

**Dataset construction**  We had a few "false starts" while constructing the dataset; many of the paired code-language dataset we initially evaluated (e.g. Kocetkov et al. (2022); Yin et al. (2018); Husain et al. (2019)) are scraped from public software repositories and often their instances cannot be executed in isolation (since that would require external libraries or modules that are not included in the dataset).

Another unsuccessful attempt was with annotating the language documentation pages directly (e.g. "reference" in docs.python.org); these contain extensive commentary but relatively few code snippets apart from the class headers and method signatures.

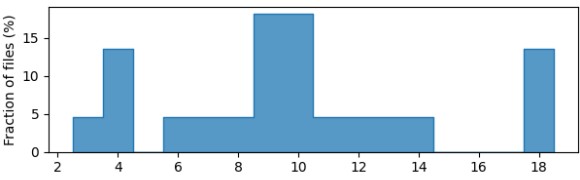

Figure 3: Annotation coverage of the dataset.

|  | min | max | mean | std.dev. |
| --- | --- | --- | --- | --- |
| Preconditions | 0 | 122 | 20.1 | 22.9 |
| Postconditions | 5 | 142 | 53.5 | 23.7 |
| Statements | 4 | 501 | 46.9 | 68.4 |

Table 1: Annotation length statistics (no. of characters)

The current version of the HTL dataset builds upon the "Programming Puzzles" dataset of Schuster et al. (2021). Each instance in the source dataset is a Python function implemented to return a Boolean indicating whether the input solves the programming puzzle or not; the code snippets in the source dataset are all on the order of a few tens of lines in length, which explains the distribution of the resulting annotations in Figure 3. Some resulting instances that appear in HTL can be seen in Figure 2.

Apart from `assert` statements, the code is mostly "referentially transparent" , i.e. it performs no file or socket I/O, however there are instances of in-place variable mutation [1].

**Annotation guidelines**  Following where possible the protocol of Dalvi et al. (2021), authors were asked to write annotations that are :

- *entailments*[2]: postconditions should immediately follow from the statement, and preconditions should be immediately precedent to it,

- *compositional*: allowing more general facts to be proven from specific ones. Considering two annotations with one span strictly containing the other one, the preconditions of the contained one can be taken to apply to the surrounding one as well.

When an annotation does not have a pre- or post-condition, we take the empty slot as shorthand for the $\top$ atom: $\{\top\}\,S\,\{Q\}$ means that whenever $S$ halts $Q$ holds; conversely, $\{P\}\,S\,\{\top\}$ is true for all compatible and terminating $P$ and $S$.

Since the annotations are free-text, we enclose in single quotes any references to program variables

---

[1]More specifically, a reference can be substituted with the definition it points to. N.B.: `assert`s are optimized away with the `-O` runtime flag, and the absence of side effects cannot be determined a priori in Python.

[2]$A$ entails $B$ (written $A \vdash B$) if $A$ cannot be true without $B$ being true.

or built-in names., e.g. `'state' has at least 'i' elements`, in order to later tokenize them as distinct entities.

**Annotation tool**  We implemented an interactive code annotation tool, consisting of a browser-based frontend and a server that samples from the memory-mapped source dataset and implements a simple REST API for annotating single triples, and proceeding to the next example (see Figure 4).

The design of the annotation UI underwent a few iterations too, in order to support a human programmer with contextual feedback and allow to build upon previous knowledge incrementally.

## 3   Related Work

**Datasets for program understanding**  Semantic parsing (SP) (i.e. translating a natural utterance to formal language while preserving meaning) and program summarization (the inverse process) are well-studied problems that progressed together with the diffusion of code-natural language datasets (such as Yu et al. (2018); Hasan et al. (2021); Lu et al. (2021); Lai et al. (2022)). Most of the larger datasets employ some form of automatic web scraping of sources that contain both code and natural-language descriptions (e.g. comments, forum questions and answers, etc.), followed by multiple steps of de-duplication and human curation; this approach trades off specificity for scale, and is in general not guaranteed to contain functional code or informative text (compare with our desiderata in Section 2).

**Reasoning with natural language**  Bos and Markert (2005) discuss the approximate entailment problem and "distance" to entailment, but their semantic representation is based on first-order logic formulas and grounded with WordNet.

Clark et al. (2021) demonstrate deductive reasoning in transformer-based language models using deductive rules expressed in a restricted natural language. In their setting, all premises and rules must be explicitly stated to construct valid training instances.

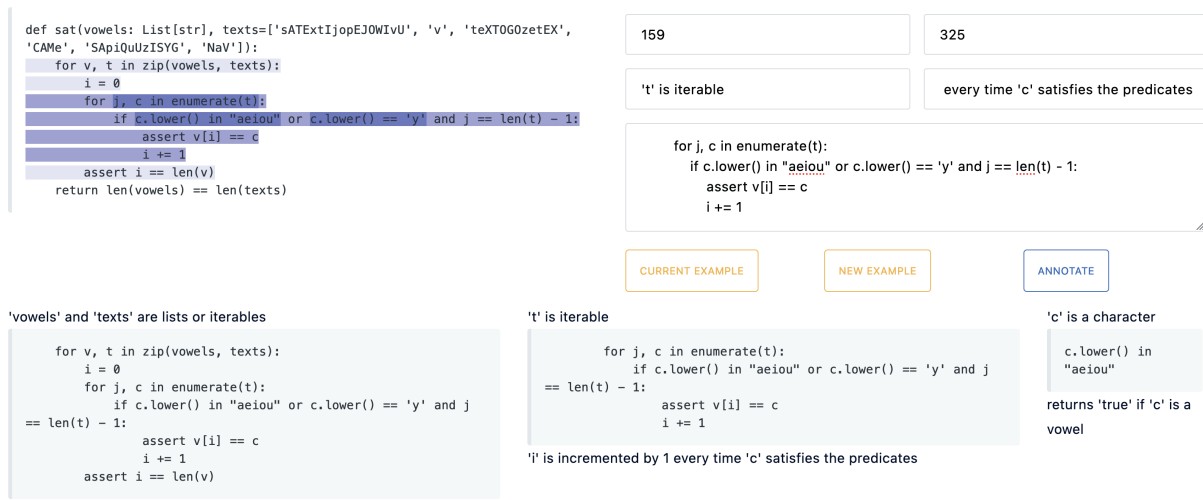

Figure 4: The user interface of the HTL data labeler. Top left : one code instance. Top right : when the user highlights a span of code, a callback copies it to the textbox, and the user can input pre- and a post-condition. After each annotation round, the updated ranges of the current annotations are superimposed on the code as a heatmap. Bottom row: as the user hovers on the highlighted spans, the backend retrieves the annotations that overlap with the cursor.

## 4 Discussion

In summary, we presented HTL , a new dataset of fine-grained annotations of program behaviour written in natural language and structured as Hoare triples. We discussed the limitations of some current labelling approaches for program understanding datasets, and presented a new annotation protocol.

This work raised more questions than it answered; we hope the dataset we presented here can serve as a foundation for addressing some of the following ones.

**Grounding** Our dataset currently lacks a connection between surface forms (program and language strings) and valuations, and only associates the two language modalities.

**Open-world vs. closed-world** Due to the lack of typing of Python code, our annotations make assumptions that might be unnecessarily restrictive; it would be interesting to follow previous work (Tafjord et al., 2021) and consider *Unknown* predicate valuations as well.

**Training language models on HTL** We conjecture that a language model can be trained with a linearized tree of triples (similarly to the encoding scheme of Dalvi et al. (2021)), e.g. by aiming to reconstruct a missing pre- or a post-condition. In symbols, one could approximate or optimize for $\tilde{Q}$

s.t. $(P, S) \vdash \tilde{Q}$ (deduction) and $\tilde{P}$ s.t. $\tilde{P} \dashv (S, Q)$ (abduction), similarly to recent "masked" language modeling objectives.

By "fine-tuning" a preexisting language model such as T5 (Raffel et al., 2020) with HTL we hope to leverage the model's implicit world knowledge and language proficiency (e.g. with regards to synonymity, polysemy etc.) which could help when aligning semantically-related annotations.

**Automatic labeling** Understanding and annotating program fragments is a very laborious and time-consuming process. Could we delegate this task to a generative language model instead?

## Limitations

Since the source dataset (Schuster et al., 2021) addresses algorithm design puzzles, it is implemented using functions that do not have computational side effects, does not define and use new types and classes, and additionally does not admit imported modules or libraries external to the Python standard library. Consequently our derived HTL dataset is subject to the same limitations and as such its instances are not representative of many real-world programs.

The annotations are written by humans; as such and despite best efforts they might be noisy and of an uneven standard across the corpus.

## Ethics Statement

Our dataset contains fine-grained annotations on the behaviour of program code; as such, the knowledge it embodies is of a very specialized nature and in particular it does not carry potentially harmful biases against any culture or society. We release our HTL dataset for research purposes only.

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

# A Annotation Schema

```
{
  "annTriple": {
    "postcondition": "declares a lambda function that returns the 'i+1'th element of 'kept'",
    "precondition": "'kept' is a list",
    "statement": {
      "trContent": "lambda i: kept[i]",
      "trEnd": 401,
      "trStart": 384
    }
  }
}
```

Figure 5: An annotated triple from the `FilterInts:0` instance

Above in Figure 5 is an example of an annotated triple, serialized as a JSON object. The fields `precondition` and `postcondition` are called $P$ and $Q$ respectively in the main text whereas the `trStart`, `trEnd` fields are the position within the original data instance of the start and end of the highlighted code span (i.e. the `trContent` field which corresponds to $S$ in the Hoare triple definition).