# OpenReview forum: "Natural Language Annotations for Reasoning about Program Semantics"
_EMNLP/2023/Conference — EMNLP 2023 Findings_

### Official Review · Reviewer_Q8NV · 2023-07-25

**Soundness:** 4

**Excitement:**

2: Mediocre: This paper makes marginal contributions (vs non-contemporaneous work), so I would rather not see it in the conference.

**Paper Topic And Main Contributions:**

The paper proposes an annotation tool and protocal to annotate program semantics with natural language.

**Questions For The Authors:**

A. The annotation in Figure 2 looks like pseudo code instead of natural language. Have you found any linguistically interesting annotations that do not look like templates?

**Reasons To Accept:**

The authors contributed a dataset and an annotation protocol.


**Reasons To Reject:**

- The motivation is unclear to me, at least from a practical engineering perspective (e.g., to build an actually useable system). As the authors pointed out in L120, it was actually unclear how such a system would scale to more general general applications that have external libraries, which are very common in real-life applications.
- There might be computational linguistic/cognitive science motivations, though, and I am not qualified to judge. I am a bit worried that, by restricting our attention to a closed environment, we are missing the opportunity of collecting data on how language can be used to describe program semantics in a wider range of context.

**Reproducibility:**

N/A: Doesn't apply, since the paper does not include empirical results.

**Reviewer Confidence:**

2: Willing to defend my evaluation, but it is fairly likely that I missed some details, didn't understand some central points, or can't be sure about the novelty of the work.

---

> ### Author Rebuttal · Authors · 2023-08-27
>
> We thank the reviewer for their valuable feedback and we will address their points in turn:
>
> ## Reasons to reject
>
> >   The motivation is unclear to me, at least from a practical engineering perspective (e.g., to build an actually useable system). As the authors pointed out in L120, it was actually unclear how such a system would scale to more general general applications that have external libraries, which are very common in real-life applications.
>
> Since we are presenting a resource paper, we err on the conservative side with our claims and only speculate that this style of dataset could be useful for deducing program behaviour in a setting similar to that of "Entailment trees". We are still quite a way off from substantiating claims that have engineering relevance.
>
> Most programming languages have a "standard library" which is in scope by default. In Python's case, this is particularly extensive, and together with the language itself already enable one to write very rich programs (with data structures, I/O etc.).
> However here we intentionally focus on a "pure" subset of the base language, i.e. on computations that have a deterministic output (i.e. independent of the background state of the program or operating system), to align with the capabilities of the Formal Verification community that focuses on Hoare logic (e.g. inferring program invariants of loops) .
>
>
> >    There might be computational linguistic/cognitive science motivations, though, and I am not qualified to judge. I am a bit worried that, by restricting our attention to a closed environment, we are missing the opportunity of collecting data on how language can be used to describe program semantics in a wider range of context.
>
> Since natural language annotations of programs do not naturally have an unambiguous interpretation, here we focus on aligning them to program fragments that do.
> Code that imports external libraries is not fundamentally different from "vanilla Python" from an interpretation standpoint; only its evaluation harness is more technically involved.  The reviewer's concerns about applicability to in-the-wild code are legitimate, but we think that addressing those would be an extension of this work.
>
>
> ## Questions
>
> > A.The annotation in Figure 2 looks like pseudo code instead of natural language. Have you found any linguistically interesting annotations that do not look like templates?
>
> Yes, sometimes, e.g. in file `Find files or url: /Users/marco/Documents/research/code-triples-lm/labeler-hs/out/AnyTriangle:3_3753893779450932199_annot.jsonl` given the expression `a != b != c != a`  we see the postcondition `'True' if 'a' and 'b' and 'c' all have distinct values`. However producing this dataset requires a balancing act between "naturalness" and faithfulness, which is why many of the annotations indeed look generated by a template.

---

### Official Review · Reviewer_9DAE · 2023-08-05

**Soundness:** 3

**Excitement:**

3: Ambivalent: It has merits (e.g., it reports state-of-the-art results, the idea is nice), but there are key weaknesses (e.g., it describes incremental work), and it can significantly benefit from another round of revision. However, I won't object to accepting it if my co-reviewers champion it.

**Paper Topic And Main Contributions:**

This paper's main contribution is a dataset containing fragments of standalone Python programs annotated with natural language preconditions and postconditions in a manner analogous to Hoare triples. The authors also contribute their annotation tool.

This resource is intended to support the development of programming assistants with improved understanding of code semantics. From the structure of the data, it appears the authors envision a system with the ability to infer an informal "mental model" of the behavior of a piece of code, and interrogate that model at various levels of granularity to provide improved consistency and/or explainability.

**Questions For The Authors:**

A: In your annotation tool, is it possible to select previous annotations (by span) and revise them?

**Reasons To Accept:**

As the authors note in their motivation, scraping open-source repositories for comments leads to a few issues with the resulting data: comments don't cover all semantic levels of detail, and open-source projects often have complex dependencies that make it difficult to execute them in order to obtain ground-truth semantics. The proposed annotation scheme addresses the former, and the latter problem is addressed (at least in the authors' HTL dataset) by choosing to annotate standalone programs. In principle, addressing these issues should make the resulting resource useful to the community.

**Reasons To Reject:**

I don't see any issues in the proposed annotation scheme fundamental enough to merit rejection. However, the dataset (from what I can tell) appears to cover only 20 programs of the source dataset's ~400 (Schuster et al., 2021). Based on the histogram in Figure 3, the dataset as a whole contains 209 annotated triples. This restricted scale limits the usefulness of the primary contribution.

**Reproducibility:**

5: Could easily reproduce the results.

**Reviewer Confidence:**

3: Pretty sure, but there's a chance I missed something. Although I have a good feel for this area in general, I did not carefully check the paper's details, e.g., the math, experimental design, or novelty.

**Typos Grammar Style And Presentation Improvements:**

There are two orphans (single trailing lines across a page boundary): one on line 131 and one on line 270.

Please include some data statistics - at least the number of annotated triples + programs. The distribution of annotated code span lengths would be nice to see.

It's a little annoying to have to download the files individually from the Zenodo interface - it would be good to also provide them as a single archive!

---

> ### Author Rebuttal · Authors · 2023-08-27
>
> We thank the reviewer for their valuable feedback and we will address their concerns in turn.
>
> ## Reasons to reject
>
> > the dataset (from what I can tell) appears to cover only 20 programs of the source dataset's ~400 (Schuster et al., 2021).
>
> We are actively working on an extended version of this dataset aiming to distribute it with the final version of the paper.
>
> ## Questions
>
> > A is it possible to select previous annotations (by span) and revise them?
>
> That's an excellent suggestion! Not possible at present; any edits are done in the JSONL annotation files by hand.
>
>
> ## Presentation improvements
>
> > There are two orphans (single trailing lines across a page boundary): one on line 131 and one on line 270.
> We will fix.
>
> > Please include some data statistics - at least the number of annotated triples + programs. The distribution of annotated code span lengths would be nice to see.
> We will add plots with more descriptive statistics on the dataset.
>
> > It's a little annoying to have to download the files individually from the Zenodo interface - it would be good to also provide them as a single archive!
> Great suggestion, we will upload a new version with a zipped archive. The dataset link we provided will always point to the latest version.

---

### Official Review · Reviewer_nCxi · 2023-08-11

**Soundness:** 3

**Excitement:**

3: Ambivalent: It has merits (e.g., it reports state-of-the-art results, the idea is nice), but there are key weaknesses (e.g., it describes incremental work), and it can significantly benefit from another round of revision. However, I won't object to accepting it if my co-reviewers champion it.

**Paper Topic And Main Contributions:**

This paper presents a new dataset of **natural-language annotations** of program semantics (in the forms of pre- and post-conditions).

**Questions For The Authors:**

A. Why 'self-contained' and 'reason locally' should be desiderata if they fundamentally limit what types of programs to annotate? As the authors claimed, "Consequently our derived HTL dataset is subject to the same limitations and as such its instances are not representative of many real-world programs.". If that is the case, what are the downstream impacts of this dataset?

B. How can HTL augment existing large-scale bi-modal datasets scraped from GitHub? The paper conjectures that a language model can be fine-tuned with HTL but does not provide even preliminary results of how such fine-tuning can lead to a better LM.

C. What are the main benefits of using natural language to annotate pre- and post-conditions if many can be easily translated to programmatic predicates? For example, in Figure 2, " 'f' is a string" can be easily translated to `type(f) == str`.

D. Could you provide more in-depth qualitative and quantitative descriptions of HTL dataset?

**Reasons To Accept:**

+ A new bi-modal dataset that fills in the gap between informal annotations (e.g., code comments) and formal annotations (e.g., programmatic predicates)

**Reasons To Reject:**

- Lack of clear motivation for building the dataset (HTL): What are the downstream impacts of this dataset?
- No validation of annotation quality: How reliable are the annotations?
- No in-depth descriptions of the annotated dataset: What are the distributions of these annotations? Are there any patterns and categories? Any artifacts or shortcuts?

**Reproducibility:**

N/A: Doesn't apply, since the paper does not include empirical results.

**Reviewer Confidence:**

3: Pretty sure, but there's a chance I missed something. Although I have a good feel for this area in general, I did not carefully check the paper's details, e.g., the math, experimental design, or novelty.

**Typos Grammar Style And Presentation Improvements:**

- Section 2, Dataset construction: The narrative can be straight to the point. Explain why you chose "Programming Puzzles" first and then exclude other alternatives.
- Figure 3: Using a pie chart would be clearer.

---

> ### Author Rebuttal · Authors · 2023-08-27
>
> We thank the reviewer for their valuable feedback and we will address their points in turn.
>
> ## Reasons to reject
>
> > Lack of clear motivation for building the dataset (HTL): What are the downstream impacts of this dataset?
>
> As we state in the introduction, one of the motivations is towards producing a model that approximates program behaviour while allowing deductive explanations for its answers. We were inspired by the work on Entailment Trees.
> We are not presently able to comment on its downstream impact, but we view this work as somewhat fundamental and as a "conversation starter".
>
> > No validation of annotation quality: How reliable are the annotations?
>
> Human-sourced annotations for this dataset require a delicate balance between "naturalness" and faithfulness to the source data, but we are not currently aware of how to quantitatively and systematically measure this. We would love to incorporate any suggestions on this issue.
>
> > No in-depth descriptions of the annotated dataset: What are the distributions of these annotations? Are there any patterns and categories? Any artifacts or shortcuts?
>
> Thank you for this suggestion. In our final version we can include more descriptive statistics on the dataset. Indeed it would be interesting to show whether they cluster in some sense.
>
> ## Questions
>
> > A. Why 'self-contained' and 'reason locally' should be desiderata if they fundamentally limit what types of programs to annotate?
>
> We deliberately restrict the scope to a "pure" fragment of the Python language, i.e. in which the evaluation of statements only depends on their input and not on some background state (I/O etc.) with the purpose of aligning (labeling) the descriptive behaviour given in the annotations with its actual behaviour without any additional assumptions.  Adding external libraries would only make the evaluation harness more complex but the program semantics wouldn't be qualitatively different. On the other hand, adding global state or I/O would also complicate the semantic model, and we see this as an extension to this work.
>
> > B. How can HTL augment existing large-scale bi-modal datasets scraped from GitHub? The paper conjectures that a language model can be fine-tuned with HTL but does not provide even preliminary results of how such fine-tuning can lead to a better LM.
>
> That's a great question which we are not prepared to answer at present. Indeed, as described in our "false starts" paragraph we initially tried with a scraped dataset but these make a lot of implicit assumptions (I/O, custom types, external libraries) which are hard to pin down exhaustively.
>
> > C. What are the main benefits of using natural language to annotate pre- and post-conditions if many can be easily translated to programmatic predicates? For example, in Figure 2, " 'f' is a string" can be easily translated to type(f) == str.
>
> Indeed, the one above can easily become a programmatic assertion, but there are cases when the annotations can provide a "shortcut" and would be rather cumbersome to evaluate even approximately (e.g. see file `Find files or url: /Users/marco/Documents/research/code-triples-lm/labeler-hs/out/BackwardsDigits:1_3680073601176606239_annot.jsonl` , first statement, postcondition: `'digits' is a dictionary where the keys are strings representing the English spelling of numbers and the values are the corresponding integers`)
>
>
> > D. Could you provide more in-depth qualitative and quantitative descriptions of HTL dataset?
>
> Yes, we will include more dataset statistics in the final version.
>
> ## Presentation improvements
>
> > Section 2, Dataset construction: The narrative can be straight to the point. Explain why you chose "Programming Puzzles" first and then exclude other alternatives.
>
> We deliberately included our "false starts" to document our process and in the hope to inform others who might be interested in constructing a similar datasets.

---

### Meta-Review · Area_Chair_uBHu · 2023-09-19

**Recommendation:** 3

**Metareview:**

The paper contributes a dataset containing fragments of Python programs annotated with natural language preconditions and postconditions.

This type of dataset is novel and fills a gap among other existing datasets.

The actual potential applications of this type of dataset are left unclear in the paper.
The dataset itself is also somewhat small, which restricts its usefulness, and lacks evaluation of annotation quality.

---

### Decision · Program_Chairs · 2023-10-07

**Decision:**

Accept-Findings

**Comment:**

The paper contributes a dataset containing fragments of Python programs annotated with natural language preconditions and postconditions.

This type of dataset is novel and fills a gap among other existing datasets.

The actual potential applications of this type of dataset are left unclear in the paper.
The dataset itself is also somewhat small, which restricts its usefulness, and lacks evaluation of annotation quality.